# Noroviruses: Evolutionary Dynamics, Epidemiology, Pathogenesis, and Vaccine Advances—A Comprehensive Review

**DOI:** 10.3390/vaccines12060590

**Published:** 2024-05-29

**Authors:** Cornelius Arome Omatola, Philip Paul Mshelbwala, Martin-Luther Oseni Okolo, Anyebe Bernard Onoja, Joseph Oyiguh Abraham, David Moses Adaji, Sunday Ocholi Samson, Therisa Ojomideju Okeme, Ruth Foluke Aminu, Monday Eneojo Akor, Gideon Ayeni, Danjuma Muhammed, Phoebe Queen Akoh, Danjuma Salisu Ibrahim, Emmanuel Edegbo, Lamidi Yusuf, Helen Ojomachenwu Ocean, Sumaila Ndah Akpala, Oiza Aishat Musa, Andrew Musa Adamu

**Affiliations:** 1Department of Microbiology, Kogi State University, Anyigba 272102, Kogi State, Nigeria; omatolac@gmail.com (C.A.O.);; 2Department of Primary Industries, Orange 2800, NSW, Australia; 3Department of Virology, University College Hospital, Ibadan 211101, Oyo State, Nigeria; 4Department of Biotechnology Science and Engineering, University of Alabama, Huntsville, AL 35899, USA; 5Department of Molecular Biology, Biotechnology, and Biochemistry, Wrocław University of Science and Technology, Wybrzeże Wyspiańskiego 29, 50-370 Wrocław, Poland; 6Department of Biological Sciences, Federal University Lokoja, Lokoja 260101, Kogi State, Nigeria; 7Department of Biochemistry, Kogi State University, Anyigba 272102, Kogi State, Nigeria; 8Epidemiology and Public Health Unit, Department of Biology, Universiti Putra, Seri Kembangan 43300, Malaysia; 9Department of Microbiology, Federal University Oye, Oye 371101, Ekiti State, Nigeria; 10Department of Microbiology, Salem University, Lokoja 260101, Kogi State, Nigeria; 11Department of Biotechnology, Federal University Lokoja, Lokoja 260101, Kogi State, Nigeria; 12Australian Institute of Tropical Health and Medicine, James Cook University, Townsville 4811, QLD, Australia; 13College of Public Health Medical and Veterinary Sciences, James Cook University, Townsville 4811, QLD, Australia; 14Centre for Tropical Biosecurity, James Cook University, Townsville 4811, QLD, Australia

**Keywords:** norovirus, evolution, pathogenesis, diarrhoea, molecular diversity, vaccine

## Abstract

Noroviruses constitute a significant aetiology of sporadic and epidemic gastroenteritis in human hosts worldwide, especially among young children, the elderly, and immunocompromised patients. The low infectious dose of the virus, protracted shedding in faeces, and the ability to persist in the environment promote viral transmission in different socioeconomic settings. Considering the substantial disease burden across healthcare and community settings and the difficulty in controlling the disease, we review aspects related to current knowledge about norovirus biology, mechanisms driving the evolutionary trends, epidemiology and molecular diversity, pathogenic mechanism, and immunity to viral infection. Additionally, we discuss the reservoir hosts, intra–inter host dynamics, and potential eco-evolutionary significance. Finally, we review norovirus vaccines in the development pipeline and further discuss the various host and pathogen factors that may complicate vaccine development.

## 1. Introduction

Noroviruses remain significant viral causes of waterborne and foodborne gastroenteritis outbreaks and epidemics across all ages worldwide [1,2]. Virus transmission is principally via the faecal–oral route and involves ingesting contaminated food or water or by direct contact with contaminated environmental reservoirs or infected persons [3]. Every year, norovirus-associated gastroenteritis accounts for approximately 212,489 diarrheal deaths, and children under five years in low- to-middle-income countries (LMICs) suffer the brunt of the diarrheal burden [4]. Infants, children, and the elderly are prone to severe symptoms of norovirus infection, while the immunocompromised patients may experience chronic diarrhoea [5]. High concentrations (10^5^–10^9^ virus particles per gram of stool) of noroviruses are shed in stool from both symptomatic and asymptomatic individuals [6] thus promoting viral ubiquity. Consequently, measures to prevent people from exposure to the enteropathogens widely in circulation in the environmental reservoirs are key to breaking transmission. Currently, no approved pharmacologic therapies are available against norovirus infections and there are no licensed vaccines to prevent the disease, partly because of the incomplete understanding of human norovirus biology [5,7]. Studies on the human norovirus mechanism of entry, a critical step in pathogenesis, have met several hurdles due to the lack of infectious molecular clones and appropriate cell culture systems for in vitro propagation.

Norwalk virus was derived from samples collected during the outbreak in Norwalk, Ohio, and it was in 1972 (4 years after the outbreak) that the Norwalk agent was described as a virus by immunoelectron microscope [8]. Subsequently, the widespread recognition of the virus-like agents that all cause similar symptoms in isolated diarrheal cases led to its several names such as small-round-structured viruses, Montgomery County virus, Snow Mountain virus, Hawaii virus, Taunton virus, Mexico virus, and Toronto viruses, courtesy of the location of discovery. The development and advancement in molecular techniques through gene cloning and nucleotide sequencing provided evidence that the genome organisation of the different strains of viruses causing diarrhoea was akin to those in the family of Caliciviridae [9]. The shared genetic similarities among the strains finally resolved the confusing nomenclature, which was previously in favour of referring to each of the strains as a Norwalk-like virus (NLV), and today, is instead known as a norovirus, as was approved in 2002 by the International Committee on Taxonomy of Viruses [10].

Noroviruses are genetically diverse groups of emerging RNA viruses, which continue to evolve in humans through both point mutations and genome recombination [11]. Of note, variations in the amino acid sequences of viral proteins associated with genomic mutations have resulted in an increased period of virus-shedding [12], evasion of pre-existing immunity associated with past infections [13], and reduced virus clearance in patients infected with heterologous strains [14]. Currently, ten genogroups (GI–GX) comprising 60 distinct P-types and 49 capsid genotypes have been reported in association with human and animal infection worldwide [15]. While several different viruses can be found co-circulating, typically a single virus causes full-blown epidemics and spreads to different countries [16]. Globally, genotype GII.4 predominates, while new variants emerge almost every 2–5 years [17,18]. Usually, the accumulation of point mutations at the epitope domain of VP1 typically causes a shift in antigenic properties, which is associated with the escape of variants from herd immunity [19]. With the emerging insights into the interplay between norovirus and their hosts’ immunity versus the improvement of a human norovirus in vitro culture system, this article provides information on the past and present knowledge on norovirus biology, evolutionary drivers, epidemiology and molecular diversity, cell tropism and pathogenic mechanisms, innate and adaptive immunity, reservoir hosts, intra–inter host dynamics and potential eco-evolutionary significance. Additionally, we review the significance of the dynamic of genetic variation on the effectiveness of candidate vaccines. We highlight the potential benefit of developing broadly effective vaccines for the control of community spreads of norovirus disease. We conclude by appraising the advances in norovirus vaccines and further note the concerns of genetic heterogeneity of human noroviruses and technical challenges facing the development that need to be addressed by future studies. 

## 2. Norovirus Biology: Proteome, Genome Structure and Organisation

The genome of a human norovirus consists of a linear, single-stranded, polyadenylated RNA of approximately 7.6 kb in length [20]. The RNA genome is encased by a nonenveloped icosahedral capsid of about 27 to 40 nm in diameter [18]. The icosahedral capsid divides into the *N*-terminal region, the *C*-terminal region, the shell (S) domain, and the protruding (P) domain. The P domain divides further into P1-1, P2, and P1-2 domains. The P2 subdomain, which projects outwardly from the capsid, contains the hypervariable region which bears the histo-blood group antigen (HBGA) binding interface [21,22]. The 5′ end of the genome is linked covalently to a viral protein genome (VPg), while the 3′ end bears the poly (A) tail [18]. The genome of human noroviruses is organised into three open reading frames (ORFs) (ORF-1, ORF-2, and ORF-3). ORF1 is the largest of them all and specifies a polyprotein that cleaves proteolytically into six non-structural proteins (NSP) that function in the replication complex (NS1–4), linking of the genome (NS5 and VPg), posttranslational processing of viral polyprotein (NS6), and RNA genome replication (NS7 and RdRp). The ORF2 and ORF3, which are translated from subgenomic RNA, specify the major viral protein (VP1) and minor viral protein (VP2), respectively [18]. In the norovirus genome, genetic recombination occurs more frequently at the region of the ORF1 and ORF2 intersection and as a result, different capsid and polymerase gene genotypes are formed. Thus, to describe the recombination status, both the polymerase (RdRp) and capsid genotypes (VP1) are used in norovirus nomenclature. In the murine norovirus genome, an additional ORF (ORF4), which overlaps the ORF2, specifies a virulence factor I involved in innate immune defence and apoptosis regulation [23]. The mature viral particle contains 90 dimers of VP1 which assemble to form hollows or cup-like structures on the surface of the icosahedral viral particle. Thus, the root name “*calici*” in the family name “Caliciviridae” is derived from the Latin word *calyx* for chalix because of the characteristic cup-like structure when viewed under the electron microscope [18]. Viral particles consist of a few copies of VP2 in the interior surface of the capsid that interact with the conserved motif in the S domain of VP1. Currently, there are eleven recognised genera in the Caliciviridae family (Figure 1) based on the amino acid sequence disparities of the complete major capsid protein (VP1) sequence [11]. Both noroviruses and sapoviruses are important aetiologies of acute diarrhoea in humans, though children with sapovirus-associated gastroenteritis generally experience milder diarrheal symptoms than those caused by norovirus. The other members of the family are frequently implicated in diarrhoea involving animal and young avian mammalian species [24].

The understanding of human norovirus genome translation and replication mechanisms has been challenging due to the lack of an appropriate cell culture system. However, the recent advances involving the development of a human norovirus replicon system, the discovery of cultivable murine noroviruses with available cell culture and reverse genetics systems, and the transient in vitro gene expression assays with transfected viral genomes [25], have enabled rapid and significant progress.

## 3. Molecular Mechanisms Driving Norovirus Evolution

Point mutations in the ORF1 and ORF2 region and genetic recombination events that generate chimeric viruses are both associated with the emergence and spread of novel norovirus strains globally [16,24,26]. Though all noroviruses employ both mechanisms to generate diversity, different genotypes may favourably emerge and persist in host populations [26]. Although there are RNA viruses for which successful vaccination does occur (e.g., polioviruses, rubeola, mumps, rubella), the control and treatment of infections mediated by RNA viruses are generally quite challenging as they can rapidly and easily generate mutants capable of evading antiviral treatment or vaccination. Therefore, understanding the molecular mechanisms driving the differences in phylodynamics is imperative not only for the development of effective modalities and intervention strategies for viral control but also for understanding both the present and future impact of norovirus disease. 

### 3.1. Point Mutation

The generation of point mutations in the norovirus RNA genome has been widely observed during long-term norovirus infection in both immunocompetent hosts [27,28,29,30] and immunocompromised individuals [12,31,32,33]. Like the other RNA viruses, the lack of proofreading repair mechanisms associated with the RNA replicates and transcriptase activity remains the critical factor driving the accumulation of point mutations [34]. During replication in RNA viruses, mutations are generated at rates estimated to be 10^−3^ to 10^−5^ per nucleotide. Specifically in the norovirus genome, mutations occur at rates ranging from 1.9 to 9.0 × 10^−3^ substitutions/nucleotide/year [35,36]. 

Genome comparative studies involving norovirus genotypes of different RNA-dependent RNA–polymerase (RdRp) fidelity have provided evidence of an inverse relationship between RdRp fidelity and strain prevalence [36]. For instance, the two norovirus genotypes, GII.4 and recombinant GII.b/GII.3, which displayed a lower RdRp fidelity than the GII.7 strain, have been more prevalent locally and worldwide [34,36]. This observation suggests that the low fidelity associated with antigenic diversity may confer a fitness advantage on the variant under the pressure of population immunity. In addition, genetic alterations in regions other than RdRp may be due to antigenic drift or shift because of the accumulation of point mutations to escape the immune response [16]. In a molecular epidemiology analysis by Chen et al. [26], more mutations were identified on the P2 sub-domain of the ORF2 region than on the RdRp region of the ORF1. A recent protein mutation analysis of isolated sequences from a population-based diarrhoea surveillance study of nearly 19 years among Chinese infants revealed that amino acid sequence alterations in the epitope P2 domain of the major capsid protein (VP1) are associated with an antigenic change in the virion capsid and evolution of the virus [26,37]. Over the years, studies have shown that changes in the GII.4 HBGA-binding specificity may contribute to the increased prevalence of GII.4 genotype in acute diarrheal cases [37]. For the P2 domain of VP1 in non-GII.4 strains such as GII.2, GII.7, and recombinant GII.b/GII.3 strains, the more fitted genotypes in the population have been shown to display a high level of diversity in the epitope P2 domain [38,39]. The emergence of novel GII.4 strains with distinct point mutations in the HBGA-binding P2 domain of VP1 may lead to antigenic changes which allow the emerging strains to avoid recognition by antibodies elicited against previously circulating strains [22]. The differential antibody recognition pattern suggests that the norovirus is evolving either by increasing its antibody binding capacity or reducing it to evade immune system recognition and subsequently escape herd immunity [22,34,40].

In immunocompetent hosts, faecal shedding of norovirus typically lasts 1 to 4 weeks following diarrheal onset [28]. However, sequencing analysis of isolates from a prolonged infection has shown that genomic mutations leading to amino acid sequence alteration caused neutralisation escape and reduced virus clearance and, ultimately, prolonged virus-shedding in patients infected with novel strains [28,29,41]. Very recently, the long-term shedding of the recombinant GII.14[P7] strain in an immunocompetent host was associated with six mutations in the genomic regions encoding the RdRp, VP1, and VP2, which occurred in a time-dependent fashion over 3 months’ of infection [28]. Intra-host emergence of antigenically distinct pandemic from the historically predominant GII.4 norovirus strains has been reported [42]. This observation likely indicates that in some individuals, viral evolution resulting from the accumulation of mutations over a long-term infection leads to relevant phenotypic variations in the virion to strategically escape herd immunity [42].

### 3.2. Genetic Recombination

The recombination of genomes is a common mechanism for generating antigenic and genetic diversity in noroviruses. The recombination events leading to antigenic shift among human noroviruses occur more often in the proximity of the ORF1 and ORF2 junction but less frequently within the viral capsid coding sequences and at the overlap of ORF2 and ORF3 [38,43]. In fact, as the genomic positive-sense (+) RNA of norovirus is transcribed into negative-sense (−) RNA replicative intermediate, the latter provides the templates for genomic and subgenomic (+) RNA transcriptions [43]. During genome recombination, the virally encoded RNA-dependent RNA polymerase (RdRp) may stall at the subgenomic promoter after initiating the synthesis of a (+) RNA strand at the promoter region of 3′ end of a (−) RNA. The attendant effect is the switching of the template towards the subgenomic (−) RNA of a co-infecting virus and the formation of a recombinant strain with a novel combination of ORF1 and ORF2/ORF3 [38]. Analogous to the genome reassortment events in influenza viruses, the recombination events between ORF1 and ORF2 allows the blend of different genome portions (structural and non-structural) during coinfections, potentially giving rise to a more genetically distant strain [44]. Importantly, the ability of the viral polymerase of norovirus to switch templates at the beginning of ORF2 is of an advantage to the virion as it can facilitate its escape from the evolutionary bottlenecks of the host immunological responses through the acquisition of a new antigenic VP1 [24]. According to Bull and White [34], genetic recombination occurring within the viral capsid ORFs may potentially alter the orientation of the capsid antigenic domains, resulting in a neutralisation escape in the presence of circulating pre-existing antibodies. Of concern is that the surface variation in the region that is exposed to immunological pressure may undermine vaccine efficacy and further promote viral fitness in a population.

Both intragenotype and intergenotypic recombination have been shown to drive norovirus evolution, contributing to the emergence of several GII recombinants and potentially non-GII strains globally [2,22]. Of note, six antigenically distinct GII.4 strains (US 1995/96, Farmington Hills 2002, Hunter 2004, Den Haag 2006b, New Orleans 2009, and Sydney 2012) account for 62–80% of all norovirus outbreaks that have been described in different pandemics of acute gastroenteritis since the late 1990s. Evidently, a recent report by Lin et al. [45] showed that homologous recombination events in the capsid gene are associated with increased norovirus virulence and genotyping mistakes in molecular surveillance investigations. Corroborating this fact, the GIIb variant, first detected in 2003 in a two-year-old male child who presented with symptoms of acute gastroenteritis in Japan, rapidly increased in the Japanese population from a prevalence of 4% in 2003–2004 to 81.5% in 2005–2006 [46], an indication that the GIIb norovirus variant was still virulent in causing illness in the country. Again, intergenotypic recombination events involving a GII.4 capsid gene and a GII.P16 polymerase gene in the full-length genome have been observed among emergent norovirus GII.4 variants [47]. Studies have shown that each of the four phylogenetically recognised clusters of GII.3 strains are associated with a distinct ORF1 genotype [48]. Furthermore, the findings that the newly emerged genetic lineages demonstrated an increased potential for genetic diversity indicate that intergenic recombination, in addition to driving norovirus evolution, may alter the replication efficiency of the virus, enhance mutational rates, and further promote the immune selective advantage. Although the accumulation of point mutations by means of the error-prone RdRp generally results in the slow generation of quasispecies in RNA viruses [34,49], genetic recombination events produce significant changes in the viral genome much more efficiently, permitting antigenic shifts, the breaching of inter-host species barrier, epidemiological fitness modifications, and the potentiating of the pathogen virulence [49]. 

## 4. Epidemiology

### 4.1. Molecular Diversity

Noroviruses are a genetically diverse group of nonenveloped, positive sense, RNA viruses in the genus Norovirus and family Caliciviridae [11]. The rapidly evolving RNA virus has ten distinct genogroups (GI–GX), which are further subdivided into 60 distinct P-types and 49 genotypes based on the differences in the nucleotide sequences of the RNA-dependent RNA polymerase (RdRp) and the amino acid gene sequences of the major viral protein shell (VP1), respectively [15]. The strains capable of infecting humans belong to genogroup GI, GII, and GIV with 9, 27, and 2 genotypes, respectively [15]. Genogroup II with multiple genotypes is the most prevalent genogroup of noroviruses reported in humans worldwide. Of the GII viruses, the highly evolving and divergent GII.4 genotype accounts for 51–79% of the norovirus disease burden [1,50]. Of note, the strains of noroviruses are genetically and antigenically unstable, giving rise to new variants in nearly every 2 to 3 years through epochal evolution [18]. Notably, different strains of the GII.4 variant, such as the GII.4 Sydney, GII.4 New Orleans, GII.4 Den Haag, GII.4 Hunter, GII.4 Farmington Hills, GII.4 US1995/1996, and GII.4 Camberwell were reported in 2012, 2009, 2006, 2004, 2002, 1995, and 1994, respectively. However, the GII.4 Sydney strain, since 2012, remains dominant across the globe and has contributed to the global increase in norovirus gastroenteritis outbreaks [51]. Genotype GII.4 subtyping into variants is based on phylogenetic clustering and the recognition of new GII.4 variants depending on if they are involved in an epidemic in at least two geographically diverse locations [18]. Globally, norovirus remains the leading aetiology of acute gastroenteritis outbreaks and for over two decades, the GII.4 viruses have dominated most of the norovirus outbreaks across the globe.

Over the past five to seven years, there have been increasing reports on the occurrence of different strains of GII.4 and non-GII.4 genotypes of noroviruses in acute gastroenteritis cases across different countries in the world [1]; however, the temporary replacement of GII.4 viruses by the non-GII.4 viruses in acute gastroenteritis outbreaks have been observed in some countries. For instance, in Ethiopia, the GII.3 genotype predominated over the GII.4 viruses in norovirus cases in 2009 [2]. In China, the emergence of the norovirus GII.P16/GII.2 genotype in early 2017 provoked a prompt seasonal increase in acute norovirus gastroenteritis cases, surpassing the rates caused by the previously predominant GII.4 strain [52]. Further, the emergence of the GII.2[P16] variant strain during the 2016–2017 epidemic season also resulted in an abrupt increase in sporadic acute gastroenteritis patients in Europe and Asia [52,53], an observation indicating that the strain could emerge as widely spread strain with the potential to cause epidemic conditions. Again, a novel GII.17 strain in China rapidly replaced the previously dominant GII.4 viruses in norovirus gastroenteritis outbreaks in 2014/2015 [54], an observation likely credited to the genetically dynamic and rapidly evolving nature of the virus, in which case, mutation and recombination events occurring frequently within and between genotypes may create the opportunities for replacement of predominant genotypes in circulation with less dominant strains.

### 4.2. Transmission 

Noroviruses are excreted in the faeces of persons who are infected with the virus, and transmission occurs via a faecal–oral route involving direct person-to-person contact, aerosolised vomitus particles, waterborne, foodborne, or environmental fomites [17]. Noroviruses are highly contagious and the explosive nature of most viral outbreaks is likely due to the low human infectious dose estimated at 18–1000 viral particles [55], prolonged shedding in faeces by both symptomatic and asymptomatic individuals [56], lack of lasting immunity [57], viral stability in the environment under a broad range of temperatures, and the ability to persist for days in the environment without inactivation [17,58]. Consistently, social mixing patterns or high population density leading to increased contact rates and crowding remain the strongest risk factors for virus-to-person transmission [59]. In a home, the presence of a symptomatic individual is a disease predictor in both susceptible children and adults [60]. Additionally, foreign travel, especially one that is associated with behavioural changes while travelling or exposure to an antigenically distinct strain, increases the risk of contracting a norovirus infection [61]. Very often, there is the continuous shedding of infectious noroviral particles even after the resolution of symptoms, and coupled with the environmental durability which facilitates viral persistence in a closed environment, the consequences include the increased risk of recurrent infections, nosocomial infections, and community outbreaks [19].

Food-borne transmission is an efficient dissemination pathway for noroviruses. The transmission of norovirus via food can occur by means of contamination from infected food handlers at the point of production during preparation and service. The findings from published norovirus outbreaks showed that foodborne transmission (362/666; 54%) and food service settings (294/830; 35%) accounted for most norovirus cases reported worldwide [62]. The commonest food vehicles for viral transmission are fresh or frozen soft fruits and vegetables, undercooked or raw seafood, and ready-to-eat foods, including salads and sandwiches, which, though they require handling, involve little or no further cooking [24]. In several norovirus outbreak investigations, foods such as leafy vegetables, raspberries, fruits, and shellfish which were irrigated with or grown in water that was contaminated with faeces and then eaten raw were implicated in the disease transmission [12,63,64,65,66,67]. Bivalve molluscs such as mussels, scallops, cockles, clams, and oysters accumulate multiple norovirus strains in their edible tissues via filter feeding [38,68], presenting opportunities for infectious human norovirus inter- and intragenotype co-infection and subsequent viral genomic recombination within the host [69]. As it has been hypothesised, bivalve molluscs by virtue of filtration of both human and animal waste could serve as vectors for the introduction of both human and different animal-derived norovirus sequences into a single host [69]. Noteworthy, food chain globalisation can increase the chance of genomic exchange between distinct polymerases and coat proteins of recombinant strains and further introduces more challenges with regards to foodborne outbreaks of norovirus recognition [19]. 

The large number of noroviruses excreted in faeces suggests that virus titres in wastewaters receiving such faecal matter are high [6]. In developing countries with poor socioeconomic conditions, not more than 28% of the wastewater is normally treated prior to the release into surface waters, a practice known to promote the environmental ubiquity of enteric viruses and transmission via aquatic pathways [70]. Even after sewage treatment, norovirus contamination of water with effluent discharges has been reported [71]. The contamination of water through malfunctioning sewage systems, sewage overflows, and runoff from polluted stormwater increases the chances of norovirus transmission [72]. Based on waterborne transmission, four major hydrological emission pathways have been conceptualised from reference [70] (Figure 2), namely the connected emissions from sewerage systems in the population accessing surface water directly or following treatment, direct emissions emanating from urban and rural population settings using hanging toilets, diffuse emissions arising from the urban and rural populations practicing open defecation, and onsite emissions from large population settings using pit latrines or septic tanks. Opportunities for human exposure to noroviruses can occur when food crops and green vegetables irrigated with wastewater and filter feeders that have concentrated norovirus in their edible tissues, are consumed raw or without proper cooking. In addition, drinking of sewage-contaminated water and recreational water exposure increases the risk of waterborne transmission of the virus [72]. Outbreaks of norovirus infection have been linked to ice cubes and a leaking air ventilation valve [73], potable water sources at camps [74], municipal water systems [75], and recreational water exposure [76].

### 4.3. Diarrheic Morbidity, Mortality, and Economic Burden

Annually, norovirus is responsible for one-fifth of all cases of acute gastroenteritis associated with diarrhoea and vomiting globally [77]. In most countries that have introduced a national rotavirus vaccination program, noroviruses have surpassed rotaviruses as the most important viral aetiology of acute gastroenteritis in children [17,50,78,79]. In both developed and developing countries, most individuals show evidence of norovirus infection before attaining adulthood [51], an observation pinpointing the endemic nature and global distribution of these viruses. The norovirus is ubiquitous and with severe outcomes, including hospital admissions and deaths, occur frequently among young children and the elderly worldwide. According to a recent world health report, about 685 million diarrheal cases in all age groups are caused by norovirus each year, amongst whom children <5 years old accounted for approximately 200 million and 50,000 diarrheal morbidity and deaths, respectively. In a community-based study in England, the incidence of acute gastroenteritis due to norovirus was 21·4 episodes per 100 person-years among children <5 years compared to 3·3 episodes per 100 person-years in older children [80]. Although children from developing countries with poor socioeconomic status suffer the highest diarrheic death burden, the ubiquitous norovirus illness with similar proportions of diarrheic disease in low-middle and high-income countries, represents a significant public health problem in all socio-economic settings [51].

Asymptomatic norovirus infections are common, especially in developing countries. Recently, the difference between the symptomatic and apparently healthy children was found to be smaller in Africa than in the developed countries (13.5% vs. 9.7%) suggesting high-level transmission among children in developing countries resulting in repeated asymptomatic infections [1]. Of note, disproportionately high mortality from diarrheal disease in low-socioeconomic settings results from various reasons, including malnutrition, poor water quality, inadequate access to healthcare, decreased diagnostic capacity, and suboptimal disease management with insufficient oral rehydration and zinc supplementation [81]. Noroviruses, characterised by a low infectious dose of 18–2800 viral particles and protracted period of faecal shedding, exhibit a high degree of contagion, and the capacity to cause explosive gastroenteritis outbreaks in semi-closed settings such as hospitals, restaurants, schools, and daycare centres, nursing homes for the elderly, military camps, and cruise ships [55,59,82,83]. The economic burden of norovirus infection and outbreak management is high. Worldwide, healthcare costs and productivity losses due to norovirus are estimated at $60 billion per year. Furthermore, the stratified analysis by the World Health Organisation region showed that the United States alone accounted for the highest cost at $23.5 billion [84]. More recently, a study estimated the yearly cost of norovirus outbreaks at $7.6 million (direct medical costs) and $165.3 million (productivity losses) [85]. According to a U.S. Centres for Disease Control and Prevention report, norovirus account for approximately 1 million childhood hospital care visits each year, and before the age of 5 years, 1 in every 160 will be hospitalised, 1 in every 40 will visit the emergency department, 1 in every 7 will visit an outpatient clinic, and 1 in every 110,000 will eventually die from norovirus [51]. 

### 4.4. Reservoir Hosts, Intra–Inter Host Dynamics, and Potential Eco-Evolutionary Significance

The genus Norovirus has a wide range of reservoir hosts, which include humans, canines, sea lions, rodents, felines, pigs, bats, mice, rats, sheep, and cattle (Figure 1). [15,24,38]. Comparative genome analysis has shown that the genomes of different genogroups of norovirus only shared 51–56% nucleotide sequence identities with one another [86]. Greater diversity was observed between genogroups when only ORF2 sequences rather than the full-length genomes were compared [38,87]. Norovirus genogroups GI, GII, GIV, GVIII, and GIX (formerly GII.15) have been reported in association with human gastroenteritis cases [15]. Genogroup GII, GIII, GV, GVI, GX, and the tentative new genogroups GNA1 and GNA2 have been detected in pigs [87], cattle and sheep [24], rats and mice [88], dogs [89], bats [83], porpoises [90] and sea lions [91], respectively. 

Studies on the inter-host transmission dynamics have shown that minor antigenic variants within genogroups with a frequency of <0.01% of the population could potentiate person–person transmission [34], an observation identifying inter-host transmission events as a potentially important selection force in generating antigenically distinct pandemic strains capable of evading herd immunity. Notwithstanding the large amounts of sequence diversity in norovirus genomes, not more than 5% nucleotide sequence differences across ORF2 in global outbreak season involving GII.4 variants have been observed [44], raising the question of where these antigenically distinct pandemic strains originate from. Indeed, the inter-host evolutionary trends of noroviruses have been very often compared to those of influenza virus [44,92]. Though, unlike noroviruses, the emergence of a new variant of influenza viruses from zoonotic sources has been clearly established [92]. However, there is currently no clear evidence that the emerging strain of norovirus could cross the species barrier because of the genetic proximity between animal and human noroviruses. Notwithstanding, the frequent detection of human-like noroviruses and novel strains in stool samples of symptomatic and asymptomatic domestic and farm animals such as pigs, pet dogs, cattle, and rhesus macaques have generated interest in the possible role of animals as potential zoonotic reservoirs for emerging norovirus strains [93,94,95,96]. Interestingly, the detection of the high prevalence of serum antibodies to bovine norovirus in veterinarians and the lower detection rate in the general population in the Netherlands [97] suggest some level of both zoonotic transmission and endemic transmission of noroviruses. Again, the study by Caddy et al. [25] among dogs in the United Kingdom indicated the occurrence of antibodies to human noroviruses which was consistent with replication in the enteric tract, the findings further buttressing the potential zoonotic transmission of noroviruses between dogs and humans. A phylodynamic investigation of norovirus transmission dynamics between humans and animals suggested some level of exposure of humans to animal norovirus [95]. In a recent sequencing analysis, a norovirus GI.3 strain detected in diarrheic chimps was found to display high nucleotide sequence identity with human norovirus strain in acute paediatric gastroenteritis cases in the same regions [98], an observation further pointing to a possibility of cross-species transmission. Furthermore, the analysis of 7804 sequences from both humans and animals in China identified the human norovirus as a reverse zoonosis pathogen since more human noroviruses were usually detected in animals than the reverse [98]. However, it is premature to regard noroviruses as zoonotic or reverse zoonotic pathogens based on the current body of evidence. Therefore, efforts aimed at intensifying and sustaining targeted surveillance for noroviruses including the sampling of cohabiting humans and animals during an outbreak situation together with an unbiased method of detection may further increase the chances of catching a trans-species transmission event [99]. Under experimental conditions, human GII.4 strains have been shown to infect gnotobiotic pigs and some nonhuman primates [100,101], raising the possibility of contact transmission of norovirus and a long-lasting effect of trigger in the gut epithelium. Again, newborn pigtail macaques orally challenged with a human norovirus GII.3 strain developed symptoms of diarrhoea, and evidence for virus replication in the intestinal enterocytes was observed [102].

### 4.5. Seasonality

Human norovirus infections and outbreaks occur more frequently in the cooler winter months spanning from November to April or May to September in countries above and below the equator, respectively. However, in countries closer to the equator, the seasonality of human norovirus illness is less marked [51]. A combination of epidemiological parameters involving the host, viral, and environment have been suggested as drivers of the seasonal pattern of norovirus disease [24]. Climatic factors such as temperature (i.e., cold and dry conditions) and low relative humidity that influence the transmission of other enteric viruses like rotavirus [103,104] which have been attributed to increased norovirus activity [60]. Host factors including fluctuations in societal behaviour, sudden rise in the hospitalisation rate as a result of comorbidity, and waning herd immunity are associated with the seasonal peaks in norovirus infections [24]. Evidently, immunity to human norovirus infection and disease is not long-lasting and cross-protection by heterologous genotypes is limited. Consequently, children under five years old are repeatedly being infected as immunity fades and heterotypic strains are encountered [60].

## 5. Pathogenesis

### 5.1. Clinical Features

Generally, acute gastroenteritis due to norovirus has a short incubation period of 24 to 48 h. Characteristically, human norovirus manifests with the acute onset of nausea, abdominal cramps, vomiting, and diarrhoea [3]. In immunocompetent individuals, the disease is typically self-limiting, and clinical symptoms usually last for 2–3 days [24]. For instance, in four gastroenteritis outbreak investigations among hospitalised psychiatric patients in Taiwan, 172 patients and 7 hospital workers were affected, of which 87.5% presented with diarrhoea, 25.5% with vomiting, 4.4% with abdominal pain, and 2.2% manifested with fever. The mean duration of clinical symptoms ranged from 1.2 to 2.8 days while in 86.4% of patients, symptoms resolved within 1 to 3 days [105]. Unusual extraintestinal pathologies, including seizures in young children, acute liver disease, and encephalopathy, have also been observed in association with norovirus infections [24]. The mechanism of the extraintestinal pathologies of human norovirus disease is not well understood; however, the finding that murine noroviruses can infect dendritic cells, which are capable of active migration between tissues and draining lymph nodes, suggests noroviruses may be utilising infected dendritic cells to facilitate extraintestinal spread [24]. Post-clinical shedding of norovirus characterised by peak viral titres ranging from 10^5^ to 10^9^ genome copies/g of faeces has been documented [106]. Individuals with medical comorbidities may experience a more severe and prolonged form of the disease. For instance, in a comparative analysis of norovirus gastroenteritis outbreaks between healthy healthcare workers and hospital patients, diarrhoea and vomiting occurred in 66% and 73%, respectively among the hospital staff while in hospital patients, diarrhoea and vomiting occurred in 85% and 56%, respectively. The median duration of norovirus illness in healthy hospital staff was 2 days, 3 days in hospital patients, and >4 days in 40% of patients ≥85 years old [107]. Additionally, an immunocompromised state and infection with GII.4 strains have been associated with more severe outcomes of norovirus gastroenteritis cases [72]. Complications of norovirus infections include electrolyte abnormalities, volume depletion, malnutrition, renal insufficiency and, in young children, benign convulsions [108].

### 5.2. Tropism

Noroviruses exhibit dual tropism as they target and infect the intestinal epithelial and nonepithelial (immune) cell types [109]. Over the years, professional antigen-presenting cells (macrophages and dendritic cells) have been recognised as major target cells for norovirus replication [109]. In a murine model, target macrophage and dendritic cells were shown to bear the cell surface receptor CD300lf which facilitates cell–norovirus interaction [110]. Supporting studies in immunodeficient mice intraperitoneally challenged with a pool of human norovirus revealed viral infiltration of macrophage-like cells found in spleens and livers [111]. Furthermore, viral entry into the lamina propria was followed by replication in the myeloid as well as in lymphoid immune cells such as T and B cells in the gut-associated lymphoid tissue [112]. Certainly, the infection of gut-associated immune cells has a significant impact on the pathogenesis of norovirus and the immunological response of the host to viral infection. Regrettably, efforts to propagate human norovirus in antigen-presenting cells have not yet been realised [109]. More recently, the chemosensory tuft epithelial cells in mice, which express the CD300lf receptor were also shown to be permissive to norovirus and the attendant tropic response towards tuft cells has been credited to the immune promotion of viral pathogenesis [113]. The murine norovirus strain (MNV-CR6) establishes persistent infection in tuft cells while the acute strain of murine norovirus (MNV-1) productively infects both the professional antigen-presenting cells, lymphoid cells (T and B cells) in vitro, and gastrointestinal lymphoid tissues in vivo [114]. This dual tropism demonstrated by murine noroviruses for intestinal epithelial and immune cells suggests that the CD300lf-expressing immune cells in the gut-associated lymphoid tissues are majorly targeted during the acute infection stage while the persistent strain MNV-6 hides in CD300lf-expressing tuft cells. In the later phase, cytokines such as IL-25 and IL-4 induce the proliferation of the tuft cells and thus promote the MNV-6 persistence [114].

Noroviruses infecting humans recognise the histo-blood group antigens (HBGAs), which are glycans on the surface of mucosal epithelial cells, through the immunogenic P2 domain of VP1 [19]. As an initial attachment factor, genetic susceptibility to a Norwalk virus infection has been positively correlated with HBGA secretor status in different populations [38,115]. For instance, Lindesmith et al. [116] showed that the recognition of the a1,2-linked fucose residues, whose cell surface expression is a function of a wild-type *FUT2* gene, accounted for susceptibility to the Norwalk virus infection in approximately 80% of the human population with a wild-type *FUT2* gene (known as secretors), while approximately 20% individuals that carry a null *FUT2* allele (i.e., non-secretors) were completely resistant. In the presence of bile acids as a cofactor, HBGA binding by certain norovirus genotypes can be enhanced [117]. Despite the high degree of norovirus infectivity and the rapid transmission rate, certain groups of individuals do not develop disease symptoms or become infected following exposure. This phenomenon has been largely linked to the expression of human HBGAs, especially the types that are controlled by the FUT2 (Secretor), FUT3 (Lewis), and ABO genes [118]. Mutations in the 1,2-fucosyltransferase (FUT2) gene which prevented HBGAs expression on the surface of intestinal cells have been associated with human resistance to norovirus infections [116,119]. Among the various circulating norovirus genotypes and genogroups, differences exist with respect to the binding affinity of viral ligand (VP1) to distinct HBGAs thus explaining the variation in the individual susceptibility patterns to specific strains of norovirus [115]. Though the HBGA secretor status does not fully explain the variations in infection distribution patterns in different populations (i.e., infected and uninfected individuals) for all strains of norovirus, an observation likely suggests additional mechanisms of immunity and area of future research. The HBGA specificities of various norovirus genotypes and genogroups have been adequately reviewed elsewhere [118]. The active replication of human norovirus has been associated with human enterocytes in the epithelial layer, as evident from the expression of viral structural and non-structural proteins, which were co-localised within the same cell [108]. The findings from an in vitro stem cell-derived intestinal enteroid cell culture system that provided evidence of human norovirus replication further supports the idea that human enterocytes are permissive target cells for norovirus [120]. Contrary to the conspicuous tropism of murine norovirus for immune cells [114,115], a more recent study by Green et al. [121] showed that the enteroendocrine cells (EECs) can be infected by a human norovirus, but several other reports have shown that enterocytes are the primary site of infection in the small intestine. The discovery of permissive EECs, which are specialised epithelial cells in the small intestine with both sensory and endocrine functions, may provide further insights into the pathogenic mechanism of human norovirus diarrhoea when fully explored.

### 5.3. Local Intestinal Infection

The pathogenic mechanisms of human norovirus infection are poorly understood because of the difficulty in cultivating the virus in the intestinal epithelial cells and the lack of an appropriate animal model that can perfectly express all aspects of human disease when challenged orally. The enteric mucosa of the small intestine is believed to be the focal point of localisation of human norovirus infection [108]. Evidently, histologic examination of intestinal biopsy samples derived from human volunteers that were infected with a GII (Hawaii; GII.1) or GI (Norwalk; GI.1) norovirus revealed an intestinal mucosa with specific histological changes, which included shortening of the microvilli, broadening and blunting of the villi, pale and enlarged mitochondria, cytoplasmic vacuolisation and intercellular oedema [115,122,123]. In addition, crypt cell hyperplasia [123] and epithelial barrier dysfunction [124] have been observed in human norovirus infection. In immunocompromised transplant patients with persistent norovirus infection, biopsies from the duodenum, jejunum, and ileum have shown the presence of capsid antigens, which were not found in the stomach or colonic tissues [125]. In a gnotobiotic pig model with a human genogroup II norovirus infection, viral capsid antigens were also detected in the intestinal biopsies [126], suggesting that norovirus may invade the intestinal epithelium. Although the study by Chan et al. [31] showed that norovirus GII.4 viruses can bind to other tissues such as the lamina propria and Brunner’s glands, but there is no other evidence that these tissues are infected. These cells express type 2 HBGAs in a non-secretor-dependent fashion, which can explain the binding, but such binding in vitro has not been associated with infection in vivo. In humans that are experimentally infected with the Norwalk agent, enterocyte changes have been observed with enzymatic alterations within the brush border of the small intestine [127]. Importantly, there was a shortening of microvilli and decreased alkaline phosphatase, sucrase, and trehalase activities, leading to steatorrhoea and carbohydrate malabsorption [127]. Nausea and vomiting manifesting in norovirus patients may be a result of delayed gastric emptying, a pathophysiologic outcome of norovirus infection that was attributed to an alteration of the gastric motor functions and/or inflammation of the pyloric junction between the stomach and intestine [115,128]. 

In addition to the physiologic and structural changes in the gut enterocyte, norovirus infection has also been described in association with inflammatory infiltration into the human’s lamina propria following infection with the Norwalk [122] and Hawaii viruses [123], an indication that the proinflammatory cytokines may be playing a role in the symptomatology of norovirus infection. Unlike in asymptomatic individuals, duodenal biopsy further revealed an increased number of intraepithelial cytotoxic CD8^+^ T cells during the 0–6 days of the onset of symptoms. According to Troeger et al. [124], the observed influx of intraepithelial cytotoxic CD8^+^ T cells during norovirus infection could result in enterocyte apoptosis following the release of perforin and granzymes. Again, the presence of neuronal alpha-synuclein in most immunocompromised children with acute norovirus gastroenteritis has been suggested to influence norovirus pathogenesis through activation of the nervous system and provocation of intestinal inflammation [129].

In recent years, advances in bovine, porcine, and murine models have been facilitating progress in our understanding of norovirus pathogenesis. For instance, in gnotobiotic pigs and calves, the inoculation with human norovirus genogroup II strains resulted in mild diarrhoea [101,126], such an ability of the animals to recapitulate mild form of norovirus disease implies they could be used to study the pathogenic mechanisms of virally induced disease [109]. In a humanised mouse, the inoculation with human norovirus failed to express viral disease; however, increased viral replication was evident in the intestinal tract and systemic sites post infection [111]. Further, the non-structural protein of the virus was detected in biopsies from distant organs such as the liver and spleen [111], an observation likely suggesting viral potential for propagation in vivo. Despite these promising results, the use of these animal models had met with several drawbacks including low levels of virus replication, high costs, inadequate reagents, and their inability to fully reproduce human disease [108,109,130]. More recently, a study by Ettayebi et al. [120] successfully propagated norovirus in vitro for the first time using cell monolayers of human intestinal epithelial cells. The confirmation of these findings in several other studies [101,131,132] and the availability of advanced molecular procedures, may further revolutionise modalities to study immune response, replication strategies, and pathogenic mechanisms of norovirus infection in the future.

## 6. Immunity to Norovirus Infection

Protective Immunity to human noroviruses is complex and not yet completely understood due to the lack of an appropriate cell culture system and the fact findings indicating that individuals are repeatedly exposed to antigenically distinct virus strains over time [17,109]. Epidemiologically, seroprevalence studies have suggested that human exposures to norovirus occur frequently, with worldwide anti-norovirus antibody prevalence rates approaching >80% by adulthood. Nevertheless, adults repeatedly display a high degree of susceptibility to noroviruses when challenged naturally and experimentally [12]. In several early human challenge studies, some norovirus-infected volunteers were found to show susceptibility to reinfection by both the homotypic and heterologous strains [133,134,135]. Again, the presence of preexisting antibodies in certain individuals did not provide complete protection from infection, except such individuals were repeatedly exposed to the homologous strain within 8 weeks to 6 months. In these studies [133,134,135], some of the individuals failed to develop long-term protective immunity after experimental challenge with human norovirus GI strain, whereas in another study [116], resistance to infection by human norovirus GI strain was correlated with an early increase in mucosal IgA. The observations that preexisting antibodies in infected volunteers did not confer protective immunity and that some individuals appeared to be resistant in spite of significant exposure suggest that certain innate host factors and acquired immunity may be driving susceptibility to norovirus infection [136]. The innate genetic susceptibility or resistance to norovirus infection has been credited to the presence of host HBGAs, cellular attachment factors needed for initiation of infection [118]. Most strains of human noroviruses preferentially bind HBGAs on the surface of the gut cells, however, they cannot bind to the surface epithelial cells of individuals who lack HBGAs as described previously [118]. 

The recognition of innate genetic resistance has contributed to the understanding of adaptive immunity in recent years. It was demonstrated that serum antibody levels measured by ELISA do not accurately predict disease susceptibility or resistance to norovirus illness [17]. On the contrary, HBGA-blocking assays that measure pre-existing serum antibodies that block norovirus binding to HBGAs have been more significant as higher concentrations of the blocking antibodies have been associated with a decreased risk of illness among secretor individuals in different human infection studies [137,138]. Importantly, this functional antibody has been suggested as a likely surrogate marker for neutralising antibodies and a correlate of protection from infection in children [137]. The studies of Atmar et al. [139] indicated that unvaccinated persons with higher levels of HBGA-blocking antibody were associated with lower risk of infection and illness. Among the vaccine recipients, antibody levels were higher, and it was less apparent a correlation between antibody levels and protection, although a post-hoc level was observed. Thus, vaccination modified the level of antibody needed for protection, much as has been reported for influenza. Though, Wang et al. [140] recently developed a promising surrogate neutralisation assay for the evaluation of norovirus vaccine at the cellular level. Furthermore, HBGA blockade studies have generated comparative indicators between the genetic evolution of human norovirus genotypes and the capacity to evade blockade antibody responses. For instance, the norovirus GII.4 strains responsible for most human norovirus outbreaks evolve every 2–3 years through genetic drift to cause major epidemics [49]. Comparative analysis of the ability of antibodies to block the epidemic GII.4 strains’ binding of HBGA showed that the existing genetic distance among different virus strains accounted for the variability in antibody recognition patterns [141]. In another study, the emergence of novel norovirus GII.17 strains, with potential for global distribution, was correlated with epitope changes targeted by HBGA-blocking antibodies [142]. Further, these findings underscore the importance of herd immunity as one critical factor influencing the evolution of norovirus strains in a population. The HBGA-blocking antibodies are of the immunoglobulins G or A class and the protective mechanisms included physical blockade of the HBGA-binding site on the virus’s protein shell [143]. Again, higher titres of IgA from the salivary mucosa prior to the virus challenge and higher titres of virus-specific memory B cells have also been correlated with decreased risks of developing norovirus illness [144]. In an immunocompromised individual with chronic norovirus gastroenteritis, the development of a strain-specific HBGA-blocking antibody resulted in the resolution of diarrhoea in the patient [145]. 

Although human norovirus-specific cell-mediated-immune (CMI) responses occur following infection, their relative importance has not been thoroughly investigated, unlike that of the antibodies and B cell responses [17,49,146]. Evidently, data from a mouse model of norovirus infection did indicate the important roles T cells play in the prevention of persistent infection and promotion of viral clearance, though no comprehensible information is available regarding norovirus-specific T-cell immunity in humans [109,146]. In a human volunteer’s study, oral infection with Norwalk virus-like particles indicated an elevated IFN-γ production and complete absence of IL-4, an observation suggesting a dominant Th1 response [147]. In another set of volunteers, infection with the GII.2 virus also resulted in a significant increase in IFN-γ and IL-2 production, which favoured the dominance of Th1 immune response that was found to cross-react against GI.1 and GII.1 virus-like particles in ex vivo assays [148]. The Th1 dominance in norovirus infection is confirmed by the findings of norovirus-specific memory and effector CD4+ and CD8^+^ T cell generation from the peripheral blood of healthy donors who experienced a significant increase in IFN-γ, IL-2, TNF-α, and granulocyte–macrophage colony-stimulating factor levels, but minimal production of IL-4 and IL-10 [149]. Similarly, Pattekar et al. [146] successfully tracked norovirus-specific CD8^+^ T cells in diverse differentiation states in human blood and intestinal tissues. Further, the abundance and widespread distribution across lymphoid and intestinal tissues may suggest that CD8^+^ T-cell correlates with protection against norovirus.

## 7. Vaccine Development

The World Health Organisation report in 2019 highlighted the urgent need to prioritise the development of a human norovirus vaccine [150]; however, the development of an effective vaccine against human norovirus infection has been very challenging. The main barrier facing the realisation of this objective is the lack of efficient and reproducible in vivo and in vitro infection models [151]. Others included the high genetic and antigenic diversity displayed by norovirus, lack of complete understanding of the characteristics of immune response that protect the host from natural norovirus infection, and lack of immunological correlates of protection [152]. In spite of all the potential hurdles to the development of an effective norovirus vaccine, the interest remains high. Currently, three different types of norovirus vaccines are in the development pipeline. These candidate vaccines comprise nonreplicating virus-like particles (VLPs), recombinant adenoviruses, and P particles [152]. The norovirus VLPs are protein structures capable of mimicking the organisation and conformation of wild-type viruses. The VP1 components of the VLPs can elicit a specific antibody response without any risk of infection when administered via the enteric and parenteral routes, possibly due to the absence of the viral RNA genome in the particle [153]. Although there are different VLP production platform, the Venezuelan equine encephalitis replicon system and baculovirus replicon system are more common approaches [152]. These widely used platforms are quite inexpensive and allow for the robust use of VLPs for norovirus vaccine development. However, these approaches have some limitations including the difficulty in removing baculovirus contaminants and the possibility of antigenic masking by the components of host-derived insect cell/baculovirus expression systems. P particles are nanoparticles from the norovirus’s polymerised protruding (P) capsid domain. This highly immunogenic capsid protein domain, which is stably and readily expressed in *E. coli*, has demonstrated HBGA-binding ability and can also trigger both the innate and adaptive arm of immune responses. However, studies have shown that the VLPs can normally elicit a more balanced Th1 and Th2 cross-reactive immune response than the P-particles [154]. The third norovirus vaccine platform involving the use of recombinant adenovirus expressing a GI1 or GII4 VP1 of noroviruses has been developed [155]. This vector-based system when intranasally administered in mice expressed capsid proteins that induce specific cellular, humoral, and mucosal immune responses. Though several human norovirus vaccines had been discontinued in preclinical stages, many other vaccine candidates have progressed to different phases of clinical trials. 

## 8. Human Norovirus Vaccine in Preclinical Development

### 8.1. Human Norovirus P Particles-Based Vaccine

The P particle is a surface antigen of norovirus capable of interacting with the host receptors much like the intact virus. The particle has been proposed as a vaccine candidate because of its high immunogenicity, ability to tolerate a wide range of temperatures and pH, and its stability as demonstrated in gnotobiotic piglets [156]. In a comparative study of the protective efficacy of human norovirus P particles and VLPs, groups of neonatal gnotobiotic pigs were intranasally challenged with preparation of VLPs or P particles obtained from GII.4 strain VA387. Another group previously infected with the same virus was used as a control group. After challenge with norovirus GII.4, the vaccinated animals demonstrated a lower risk of developing gastroenteritis compared to those with natural infection, albeit the risk of disease was slightly higher in VLP recipients (46.7%) compared to those given P particles (60%). The norovirus P particle induces both the innate and adaptive immune responses, which provide cross-variant protection against human GII.4 norovirus diarrhoea in gnotobiotic piglets. Relative to the VLPs based vaccine, the P particle vaccine evoked a stronger immunological response, including significantly higher numbers of activated CD4^+^ T cells in all tissues, duodenal IFN-γ CD8^+^ T cells, T-regulatory cells in the blood, and TGF- CD4^+^ CD25^−^ FoxP3^+^ Tregs in the spleen. Although, the P particle-based vaccine is promising, it is yet to be investigated in field-based clinical efficacy studies.

### 8.2. Combined Vaccines

There is a growing interest in developing human norovirus vaccines to be administered in combination with other immunogens. A trivalent combination vaccine comprising two norovirus VLPs (GII.4-1999 and GI.3) and a rotavirus VP6 was developed to provide a broad heterotypic immunity against both norovirus and rotavirus currently responsible for most severe acute gastroenteritis in children [157]. Findings from in vitro studies have shown that the trivalent vaccine could elicit high levels of norovirus and rotavirus type specific serum antibodies with >50% avidity and intestinal antibodies [158]. The presence of rotavirus VP6 in the vaccine provided an adjuvant effect by promoting norovirus VLP uptake by the antigen-presenting cells (APCs) and improving APCs activation and maturation [159]. Similarly, a combined vaccine effective against norovirus and enterovirus 71 (EV71) is in the preclinical stage of development. In a mouse model, the immunogenicity of the bivalent vaccine containing GII.4 and EV71 VLPs was compared with monovalent GII.4 and EV71 VLPs [160]. In the study, the immune response to the bivalent VLPs based vaccine was similar to the ones induced by either of the monovalent vaccines, without evidence of immune response interference between the two antigens. Importantly, infection by EV71 was averted in a similar number of experimented mice and the inhibition of GII.4-VLP interaction with mucin did not change [160], findings suggesting a balanced antibody response between the two modalities and potential applicability of the combination strategies for preventing simultaneous infection when fully developed. 

## 9. Human Norovirus Vaccines in Clinical Stages of Development

### 9.1. Monovalent Vaccines

A GI.1 VLP vaccine adjuvanted with monophosphoryl lipid A was found to elicit norovirus specific serum antibodies in most of the intranasally challenged human recipients. Interestingly, the incidence of norovirus infection and acute gastroenteritis development was significantly reduced in vaccinated subjects compared to the control groups (incidence of infection: 61% of vaccinees vs. 82% of placebo recipients; disease: 37% of vaccinees vs. 69% of placebo recipients) [161]. The confirmation of the immunogenicity of the monovalent vaccine in human adult volunteers further highlights its potential usefulness in the prevention of norovirus infection [162]. Most of the circulating serum antibodies norovirus antigen-specific cells bear markers suggestive of homing to mucosal tissues alone or to lymphoid and mucosal tissues. Of note, the adjuvanted virus-like particle vaccine elicited a strong antigen-specific B memory immune response in a dose-dependent manner. Overall, the existence of pre-challenge norovirus specific antibodies capable of blocking the binding of VLP to its corresponding HBGA correlated with protection against norovirus infection and disease development. Nevertheless, the realisation of the genetic variability and evolutionary dynamic nature of noroviruses coupled with the lack of intergenotype cross protective clinical immunity may be driving the recent research focus on a combination vaccine.

### 9.2. Bivalent Vaccines

Bivalent vaccines containing human norovirus GI and GII VLPs are in the development pipeline. To motivate future research, an animal model was challenged intranasally with the bivalent norovirus vaccine formulated in an in situ gelling dry powder, to ascertain the safety, immunogenicity, and potential antigenic interference. The formulation was found to be safe and immunogenic as both systemic and mucosal immune responses directed against each of the VLPs increased in a dose-dependent fashion. In addition, a boosting effect of the GI and GII VLPs based vaccine was noted sequel to a second dose without immune interference [163]. In another study, recombinant VLPs of five norovirus GI Strains (GI.1-1968, GI.1-2001, GI.2-1999, GI.3-1999, and GI.4-2000) were evaluated to study the pattern of immune responses following intramuscular challenge. Peripheral blood mononuclear cells were obtained from ten volunteers infected by GI.1-1968. After stimulation with different antigenic components of the VLPs, the IFN levels from PBMCs were measured. Sixty percent of the vaccinated individuals responded to at least one GI VLPs, with only two volunteers responding to GI.1 VLPs. In the cross-reactivity studies, four in every five participants responded more robustly to other GI VLPs [164]. In a Phase I/II clinical trial study, two intramuscular doses of each vaccine were administered at an interval of ≥28 days. Afterwards, vaccinated subjects were challenged with a heterologous GII.4 strain and then observed for reduction of gastroenteritis. Interestingly, fewer numbers of the vaccinated subjects (*n* = 56) than placebo recipients (*n* = 48) manifested with vomiting and/or diarrhoea of any severity after intramuscular challenge (20% vs. 41.7%; 52% reduction; *p =* 0.028). The intramuscular inoculation of a bivalent vaccine comprising a GI.1 and a GII.4 VLP plus an adjuvant followed by a challenge with the human norovirus GII.4 resulted in a significant reduction in the incidence of infection and severity of the disease in the vaccinated population compared to the placebo group [165].

In another human subject study developed by Vaxart incorporation, a replication-defective adenovirus vector-based human norovirus vaccine expressing the GI.1 and GII.4 capsid protein has successfully completed a Phase Ib clinical trials. Interestingly, findings from the phase Ib study showed that the oral bivalent vaccine satisfied all primary and secondary endpoints for vaccine safety and immunogenicity. Further, the two antigenic components of the vaccine (human norovirus GI.1 and GII.4) were found to elicit a robust mucosal immunological response in most of the recipients without evidence of immune interference [99,166]. Another promising vaccine in the development pipeline is Takeda’s bivalent (GI.1 and GII.4) VLP vaccine which is administered via the intramuscular route. The clinical phase II trial results showed that the candidate Takeda’s VLP vaccines were well-tolerated and provoked a robust immune response in healthy infants and children [167]. A clinical phase II trial in healthy adults and the elderly did not raise any safety concerns, and a high antibody response was induced by the vaccine, also observed in younger children [168,169]. Additional clinical phase II studies are ongoing to evaluate the long-term immunogenicity in adults and the vaccine efficacy in military recruits [151]. The intranasally and intramuscularly administered VLP vaccine by Ligocyte Pharmaceuticals [163] and Takeda [167] in its early and later stages of development, respectively, is now being developed by a clinical-stage biopharmaceutical company called Hillevax [170]. In a dose-finding phase 2 randomised, a double-blind trial of the HilleVax bivalent VLP vaccine candidate (HIL-214) in two cohorts of children aged 6–≤12 months and 1–≤4 years, all the vaccine formulations were found to be well tolerated and no adverse event was recorded [170]. Findings related to HIL-214 intake suggest it is a promising vaccine candidate for protecting susceptible young children against norovirus when fully developed. HilleVax has announced the completion of enrolment of over 3000 subjects in six countries (Columbia, United States, Mexico, Dominican Republic, Panama, and Peru) for the Norovirus Efficacy and Safety Trial for Infants (NEST-IN1) in 2024 (http://www.hillevax.com/ accessed on 2 February 2024). Together, the available VLP-based norovirus vaccines have shown great promise [64,159,163,164,165,167,168,169,170,171,172,173,174,175,176] (Table 1), but concerns remain about the following: (1) the efficacy in the community against commonly circulating and genetically heterogenous viruses whose pandemic strains are easily replaced within short time intervals [12]. This factor may necessitate frequent vaccine reformulations in response to evolution as it is currently in place for influenza viruses. Besides, several findings on the lack of intergenogroup cross-protection in human challenge studies [132] may suggest the need for incorporation of multiple norovirus strains in vaccine preparation for effective immunity to be elicited against each genogroup; (2) whether the magnitude or duration of protective immune response elicited by the nonreplicating antigens will be long enough to be clinically relevant and finally; (3) whether the vaccine will protect more vulnerable groups such as children and the elderly or whether the same formulations will be effective for different populations (high-income versus low income) [12,109,115].

## 10. Conclusions and Perspectives

Norovirus gastroenteritis still greatly threatens public health in all age groups because of its global societal and economic burden. In most countries that have introduced rotavirus vaccination into national immunisation schedules, norovirus has become the most significant viral cause of acute gastroenteritis. The rising trend in the burden of norovirus indicates that improved hygiene and sanitation alone may not be fully effective in controlling the spread and distribution of the disease. For decades, vaccines and therapeutics against noroviruses have been a recognised public health need; however, their development has been difficult due to the incomplete understanding of norovirus pathogenesis in the human host. Interestingly, recent advances in understanding the interplay among the viral strains, HBGAs types, and host susceptibility to diseases have provided the basis for new insights into cell tropism, an understanding of the interaction among various host and viral attachment factors as well as the elucidation of different virus-like protein morphologies. Even though the advances are yet to be extended to clinical practice, they have significantly facilitated progress in our understanding of how norovirus causes disease and persists in different populations worldwide. Again, recent findings from in vivo and in vitro studies have increased our understanding of the interplay of norovirus and host immunity. However, the interplay between the duo still poses several unanswered questions as highlighted in the immunity and vaccine section. Further, successes in culturing human norovirus have potentially added to our understanding of the biology and molecular mechanisms of the virus and further inspired new areas of research targeting the control methods. Notwithstanding, there is the need to develop and optimise these methods further to realise the objective of robust levels of replication [12]. Sequel to human exposure, norovirus competition for interaction with the host receptor during entry or cellular machinery post-penetration is a naturally occurring physiological process that cannot be altered by any simple approach. Thus, in addition to the available multi-barrier measures to prevent norovirus infection via water quality and sanitation improvement, the current efforts to develop effective vaccines should be strengthened. Interestingly, vaccine development against norovirus has progressed to preclinical phases and even clinical trial testing of candidates. In view of the existing evidence and concerns that the lack of intergenogroup cross-protection and non-lasting human immunity to norovirus could undermine current efforts targeted at developing an effective vaccine against the virus, overcoming these challenges may ultimately determine the fate of a norovirus vaccine. The inherited host variability, genetic diversity displayed by noroviruses, and ongoing viral evolution are important factors complicating the process of current vaccine development. In addition, the identification of the most suitable target age group remains to be addressed in norovirus vaccine development as most human-based trials enrolled adults rather than children in the study. Thus, it is has become necessary to know whether the candidate vaccines are effective in all age groups, including younger children who bear the brunt of the diarrheic burden. Importantly, the ability of the currently available vaccines to protect against heterologous norovirus strains and the antigenic variants that emerge frequently to cause outbreaks must be well defined by future norovirus research. The process of achieving these goals may be facilitated through the identification of a suitable immunological correlation of protection against norovirus, development of more permissive cell lines for viral culture and the availability of in vivo infection models that can fully recapitulate human disease. Areas of future norovirus research may overcome technical limitations, such as the inability to efficiently cultivate norovirus in vitro or develop a broadly effective norovirus vaccine.

## Figures and Tables

**Figure 1 vaccines-12-00590-f001:**
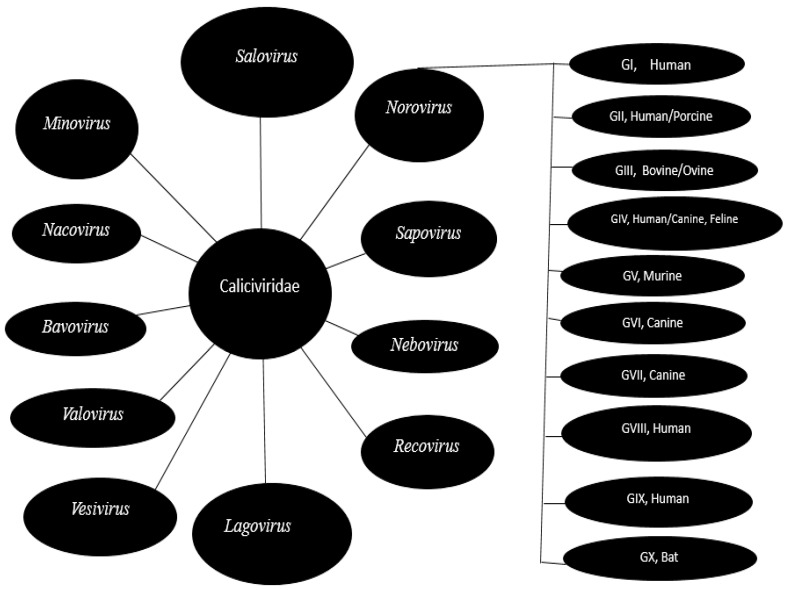
Norovirus genera within the Caliciviridae family. The established genogroups are designated GI–GX beside their corresponding host species of origin.

**Figure 2 vaccines-12-00590-f002:**
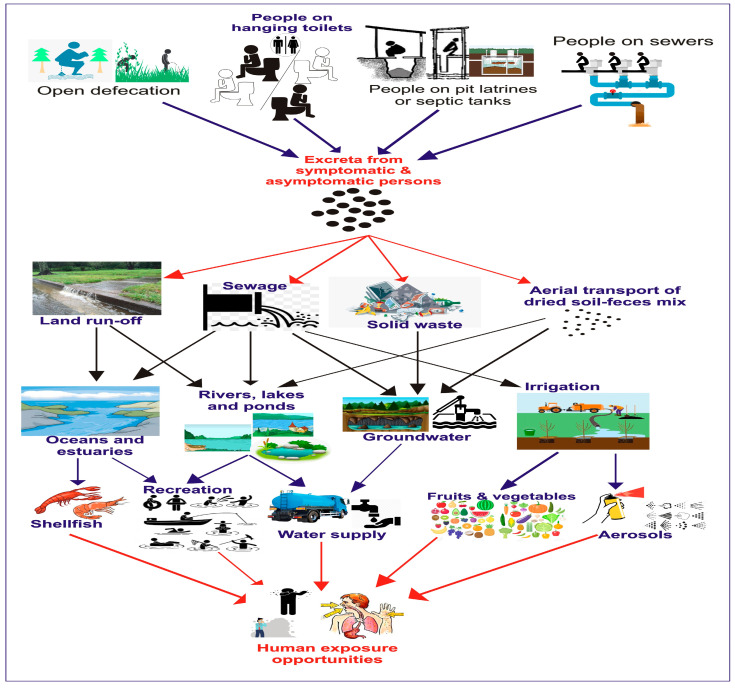
Norovirus emission pathways conceptualized from Ref. [70].

**Table 1 vaccines-12-00590-t001:** Major human norovirus vaccines in development pipelines.

Vaccine Type	Vaccine Antigens	Expression System	Development Stage	Ref.
Virus-like particles (VLP)	GII.4	*Nicotiana benthamiana*	Preclinical	[172]
	GI, GII.4, Rotavirus VP6	*Baculovirus*	Preclinical	[159,171]
	GI.1	*Baculovirus*	Clinical Phase 2a	[161]
	GI.1, GII.4	*Baculovirus*	Clinical Phase 2b	[163,167,168,169,170,174]
	GI.1, GII.4	*Hansenula polymorpha*	Clinical Phase 1	[175]
	GI.1, GII.3, GII.4, GII.17	*Pichia pastoris*	Clinical Phase 1/IIa ongoing	[176]
Recombinant Adenovirus				
	GI and/or GII.4	*Adenovirus*	Clinical Phase 1	[99,166,173]
	GI.1	*Adenovirus*	Clinical Phase 1	[64]

## Data Availability

Not applicable.

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
