# Peer review of "Noroviruses: Evolutionary Dynamics, Epidemiology, Pathogenesis, and Vaccine Advances—A Comprehensive Review"

_vaccines, 2024, doi:10.3390/vaccines12060590_

Round 1

Reviewer 1 Report

Comments and Suggestions for Authors

Omatola et al. prepared a comprehensive review of the evolutionary dynamics, epidemiology, pathogenesis, and vaccine advances for noroviruses. The English language usage is clear and professional, and the manuscript is overall easy to follow. Although this is a lengthy review manuscript, the subtitles used are accurate and straightforward, making the manuscript easy to read and for readers to refer to the section of interest. I therefore recommend it to be published in Vaccines after considering the following suggestions and making necessary revisions:

(1) In the title, the review is described as "critical". It is recommended either to describe in the abstract or conclusion section how this review would be critical, or consider using "comprehensive" instead.

(2) It is recommended in the "conclusion and perspective" section to include discussions of the limitations of current vaccines and the gap in noroviruses research field.

(3) In the section "Norovirus biology: proteome, genome structure and organisation", it would be helpful to include a Figure or Scheme to briefly show the structure (domains, 5' end, 3' end polyA, ORF, etc.), so the readers can better visualize it. This is a recommendation but not required.

(4) Figure 2: the figure seems to be overstretched. Also, please explain why different colors of arrows were used. If not necessary, it is recommended to use monochrome arrows.

Author Response

Omatola et al. prepared a comprehensive review of the evolutionary dynamics, epidemiology, pathogenesis, and vaccine advances for noroviruses. The English language usage is clear and professional, and the manuscript is overall easy to follow. Although this is a lengthy review manuscript, the subtitles used are accurate and straightforward, making the manuscript easy to read and for readers to refer to the section of interest. I therefore recommend it to be published in Vaccines after considering the following suggestions and making necessary revisions:

Author Response: Authors appreciate the reviewer for the quality of the review and the many suggestions which have improved our article.

(1) In the title, the review is described as "critical". It is recommended either to describe in the abstract or conclusion section how this review would be critical, or consider using "comprehensive" instead.

Author Response: Critical has been replaced with Comprehensive, thank you. (Title page)

(2) It is recommended in the "conclusion and perspective" section to include discussions of the limitations of current vaccines and the gap in noroviruses research field.

Author Response: Done. Concluding section, Page 19-20

(3) In the section "Norovirus biology: proteome, genome structure and organisation", it would be helpful to include a Figure or Scheme to briefly show the structure (domains, 5' end, 3' end polyA, ORF, etc.), so the readers can better visualize it. This is a recommendation but not required.

Author Response: Authors consider it fine without the suggested figure since it could be found in the work of several authors cited in the section.

(4) Figure 2: the figure seems to be overstretched. Also, please explain why different colors of arrows were used. If not necessary, it is recommended to use monochrome arrows.

Author ResponseWe appreciate the reviewer for his observation. Figure 2 was conceptualized to convey information related to several norovirus transmission pathways to the readers. Thus, the reason it appears to be overstretched, to ensures it adequately convey intended lessons to readers.

Authors have used the monochrome arrows as suggested by the reviewer (Figure 2)

Reviewer 2 Report

Comments and Suggestions for Authors

The review paper by Omatola, et al provides a comprehensive summary of the evolutionary dynamics, epidemiology, pathogenesis, and vaccine advancements of noroviruses. Norovirus-induced gastroenteritis continues to pose a significant threat to global public health, particularly in low- to middle-income countries. However, due to the complex classification and mixed information available on norovirus, there exists an incomplete understanding of this virus. Therefore, it is important to publish a comprehensive and updated review that facilitates quick and easy comprehension of this virus. This manuscript thoroughly reviews the biological features, mechanisms of evolution, epidemiology, and current advances in vaccines for noroviruses. Overall, this manuscript is suitable for publication in Vaccines. There are some comments below for author’s consideration:

1 The first part “Historical perspectives and etymology”, provides limited information; therefore, authors may consider merging this section into the Introduction.

2 Authors may introduce the “Molecular diversity” (line 359) at the beginning of the “Epidemiology” section for better coherence.

3 To enhance clarity, it is recommended to move the "Clinical features" (line 586) to the beginning of the "Pathogenesis" section. Additionally, an overview figure of the “Pathogenesis” section could help readers understand this context.

4 Following the "Vaccine development" section, authors should introduce "Human norovirus vaccine in preclinical development." First.

5 Figure 1 requires embellishment to improve its visual appeal.

6 Figure 2 seems compressed and needs to be adjusted

7 Check the format in line 546 and 792.

Author Response

The review paper by Omatola, et al provides a comprehensive summary of the evolutionary dynamics, epidemiology, pathogenesis, and vaccine advancements of noroviruses. Norovirus-induced gastroenteritis continues to pose a significant threat to global public health, particularly in low- to middle-income countries. However, due to the complex classification and mixed information available on norovirus, there exists an incomplete understanding of this virus. Therefore, it is important to publish a comprehensive and updated review that facilitates quick and easy comprehension of this virus. This manuscript thoroughly reviews the biological features, mechanisms of evolution, epidemiology, and current advances in vaccines for noroviruses. Overall, this manuscript is suitable for publication in Vaccines. There are some comments below for author’s consideration:

Author ResponseAuthors appreciate the reviewer for the quality of the review and the many suggestions which have improved our article.

1 The first part “Historical perspectives and etymology”, provides limited information; therefore, authors may consider merging this section into the Introduction.

Author ResponseThe “historical perspectives and etymology” section has been collapsed with the introduction section based on the justification provided by the reviewer

Highlighted section on page 2, Introductory section

2 Authors may introduce the “Molecular diversity” (line 359) at the beginning of the “Epidemiology” section for better coherence.

Author Response: Done, thank you. Highlighted section on page 6-7, Epidemiology section

3 To enhance clarity, it is recommended to move the "Clinical features" (line 586) to the beginning of the "Pathogenesis" section. Additionally, an overview figure of the “Pathogenesis” section could help readers understand this context.

Author Response: The clinical features has been moved as suggested by the reviewer. However, authors feel the information relating to the pathogenesis is easy to understand in its current form without supporting it with an overview figure. Besides, this section of pathogenesis is still poorly understood as highlighted in the manuscript.

For clinical feature, see the highlighted section on page 11-12, pathogenesis section

4 Following the "Vaccine development" section, authors should introduce "Human norovirus vaccine in preclinical development." First.

Author Response: Done, thank you

5 Figure 1 requires embellishment to improve its visual appeal.

Author Response: Done. Figure 1, on page 4

6 Figure 2 seems compressed and needs to be adjusted

Author Response: Done. Figure 2, on page 9

7 Check the format in line 546 and 792.

Author Response: The format has been unified throughout